# Polymorphic Variants of V-Maf Musculoaponeurotic Fibrosarcoma Oncogene Homolog B (rs13041247 and rs11696257) and Risk of Non-Syndromic Cleft Lip/Palate: Systematic Review and Meta-Analysis

**DOI:** 10.3390/ijerph16152792

**Published:** 2019-08-05

**Authors:** Mohammad Moslem Imani, Pia Lopez-Jornet, Eduardo Pons-Fuster López, Masoud Sadeghi

**Affiliations:** 1Department of Orthodontics, School of Dentistry, Kermanshah University of Medical Sciences, Kermanshah 6713954658, Iran; 2Facultad de Medicina y Odontologia Universidad de Murcia, Hospital Morales Meseguer, Clinica Odontologic Adv Marques Velez s/n, 30008 Murcia, Spain; 3Insitituto Murciano de Investigación Biomédica, Murcia, Campus de Ciencias de la Salud, Carretera Buenavista s/n, El Palmar, 30120 Murcia, Spain; 4Medical Biology Research Center, Kermanshah University of Medical Sciences, Kermanshah 6714415185, Iran; 5Students Research Committee, Kermanshah University of Medical Sciences, Kermanshah 6715847141, Iran

**Keywords:** oral cleft, non-syndromic cleft lip/palate, polymorphism, MAFB, meta-analysis

## Abstract

*Background:* Non-syndromic cleft lip/palate (NSCL/P) has an etiology, including both genetic and environmental factors. Herein, we evaluated the association of rs13041247 and rs11696257 v-maf musculoaponeurotic fibrosarcoma oncogene homolog B (*MAFB*) polymorphisms with the risk of NSCL/P in a meta-analysis. *Methods:* The PubMed/Medline, Scopus, Cochrane Library, Web of Science, and HuGE Navigator databases were systematically searched to retrieve relevant articles published up to January 2019. The Newcastle–Ottawa scale was applied for quality evaluation of retrieved articles. The 95% confidence interval (CI) and crude odds ratio (OR) were calculated for each study using the Review Manager 5.3 software to show the association between *MAFB* polymorphisms and risk of NSCL/P. The comprehensive meta-analysis 2.0 software was used to calculate the publication bias. In addition, sensitivity analysis was carried out to show the stability of results. *Results:* Of 102 articles retrieved from the databases, 10 articles were analyzed in this meta-analysis. Ten articles, including eleven studies reporting rs13041247 *MAFB* polymorphism, included 3082 NSCL/P patients and 4104 controls. Three studies that reported rs11696257 *MAFB* polymorphism involved 845 NSCL/P patients and 927 controls. The rs11696257 *MAFB* polymorphism was not associated with the risk of NSCL/P, but the CC and TC genotypes of rs13041247 polymorphism were associated with the risk of NSCL/P. Nevertheless, the C allele and CC and TC genotypes were associated with a significant decline in the risk of NSCL/P in population-based studies. *Conclusions:* The results of this meta-analysis demonstrated that the risk of NSCL/P was related to rs13041247 polymorphism, not rs11696257 *MAFB* polymorphism. Well-designed studies are required to assess the interaction of *MAFB* and other genes with environmental factors in different ethnic groups.

## 1. Introduction

The interaction between genetic and environmental factors may result in the development of non-syndromic cleft lip/palate (NSCL/P), which is the most common human congenital anomaly worldwide [1,2]. This disorder has an etiology with several factors that involves both genetic and environmental factors [3]. The live-birth prevalence of NSCL/P changes by the geographical region and ethnicity (1/2500 in African, 1/1000 in European, and 1/500 in Asian and American Indian populations) [4]. However, the NSCL/P etiology is unknown, and its underlying molecular mechanisms remain poorly understood, but some recent meta-analyses showed the association of genetics with the development of NSCL/P [2,5,6,7]. The 20q12 locus includes the V-Maf musculoaponeurotic fibrosarcoma oncogene homolog B (*MAFB*) encoded by the *MAFB* gene [8]. This gene is a transcription factor that acts as an important regulator during the development of brain structures, endocrine cells, and hematopoietic system [9] as well as the orofacial development [10,11,12]. The gene belongs to the *MAF* family of transcription factors characterized by a typical basic leucine zipper (bZip) structure [10,11,12] that is a putative tumor suppressor in the myeloid lineage [12]. It has a key function in monopoiesis [12] as well as in monocyte-dendritic cell differentiation [8]. Assessment of the genes involved in craniofacial disorders is an appropriate approach to support the genetic architecture discovery of the facial morphology [3]. In addition, findings also showed different roles of these two *MAFB* polymorphisms in the development of NSCL/P in the Asian population [13]. Therefore, this meta-analysis investigated the association of two important variants of 20q12 chromosome, including rs13041247 and rs11696257 *MAFB* polymorphisms with the risk of NSCL/P.

## 2. Materials and Methods

This meta-analysis is reported according to the PRISMA guidance [14].

### 2.1. Search Strategy

The PubMed/Medline, Scopus, Cochrane Library, Web of Science, and HuGE Navigator databases were systematically searched to retrieve relevant articles published up to January 2019. The searched terms were: (*MAFB* or v-maf musculoaponeurotic fibrosarcoma oncogene homolog B) and (“orofacial cleft”, “cleft lip”, “cleft palate”, or “oral cleft”) with no language restriction. In addition, we investigated the references of the retrieved articles related to the topic to make sure no article was missed.

### 2.2. Study Selection

The first author (M.S) searched for articles in the databases and evaluated the titles and abstracts of the relevant articles, and then, the full-texts of the articles that met the eligibility criteria were uploaded and screened. The reason for the exclusion of any article excluded after full-text screening was reported. The second author (M.M.I) independently re-assessed the relevant articles. If there was a disagreement between the two authors, the problem was resolved by discussion.

### 2.3. Eligibility Criteria

The inclusion criteria were: (1) Human and unrelated case-control studies, (2) studies including NSCL/P patients with no other systematic disease and healthy controls, (3) studies reporting rs13041247 and/or rs11696257 *MAFB* polymorphisms, and (4) studies having the sufficient data to calculate the odds ratios (ORs) and 95% confidence intervals (CIs). The exclusion criteria were: (1) Animal studies, family-based studies, the studies reporting parents who have children with NSCL/P as cases and/or parents without NSCL/P children as controls, erratum studies, case reports, review articles, systematic reviews, and conference papers and (2) studies without having the required data to calculate the genotype distributions.

### 2.4. Data Extraction

The first authors’ name, publication year, ethnicity, controls’ source, number of NSCL/P patients and controls based on each genotype and related polymorphisms, genotyping method, *p*-value of the Hardy–Weinberg equilibrium (HWE) in controls, and the quality score were extracted from the articles included in this meta-analysis. Data extraction was independently performed by two authors (M.S and P.L.-J).

### 2.5. Quality Assessment

The Newcastle–Ottawa scale (NOS) was applied for the quality assessment of articles with a maximum score of nine [15]. The quality of each study was high if score ≥ 7 was obtained. Quality evaluation was done by one of the authors (M.S).

### 2.6. Qualitative Synthesis

The qualitative synthesis, which includes assessment of bias and heterogeneity, describes the characteristics and findings of retrieved studies based on the extracted data. The estimated effects with CIs for each study analyzed in the meta-analysis were reported based on five genetic models [16] to have a more complete analysis, but this raises an issue of multiplicity of testing.

### 2.7. Statistical Analysis

The 95% CI and crude OR for each study were obtained by Review Manager 5.3 (RevMan 5.3) to present the association between the *MAFB* polymorphisms and risk of NSCL/P. The pooled OR significance was shown by the Z test, with a *p*-value < 0.05. The Chi^2^ test, the Tau^2^, and the I^2^ statistic were used to estimate heterogeneity between the studies and *p*-value < 0.1 (I^2^ > 50%) presented statistically significant heterogeneity. If there was heterogeneity, the random-effects model was conducted to allow for probable differences in the results of studies (DerSimonian and Laird method) [17]. Otherwise, the fixed-effects model was used (Mantel–Haenszel method) [18]. In addition, the HWE was calculated by the Chi-square test for the controls in each study.

The subgroup analysis was done based on ethnicity, controls’ source, and genotyping method for rs13041247 *MAFB* polymorphism. We used the Comprehensive Meta-Analysis version 2.0 (CMA 2.0) software (Biostat Inc., Englewood, NJ, USA) to perform the funnel plot analysis [19] using both Egger’s and Begg’s tests (*p* < 0.05 (two-tailed) was considered as significant existence of publication bias). In addition, cumulative analysis was done after excluding one study to assess the stability of the results. If it did not change the pooled data, the stability or consistency of the analyses would be confirmed. The meta-regression as a possible source of heterogeneity was used with the *p*-value and regression coefficient (*r*) to assess the strength of the association among the study period, the sample size, and *p*-value of the HWE in controls with the risk of the rs13041247 polymorphism.

## 3. Results

### 3.1. Study Selection

One hundred and two articles were retrieved from the databases. After removing the duplicates, 35 articles were screened (Figure 1). Of 35 articles, 20 articles were excluded based on their title/abstract because they were not related to the subject. Full-texts of 15 articles were assessed for eligibility, out of which, five articles were excluded with reasons (one article was erratum, two articles were family-based studies, one article evaluated cleft palate only, one article evaluated NSCL/P patients with tooth agenesis). Finally, 10 articles including eleven studies were analyzed in this meta-analysis.

### 3.2. Study Characteristics

Some basic features of eight articles reviewed in this meta-analysis are presented in Table 1. The studies were published from 2011 to 2015. Four articles [13,20,21,22] reported polymorphisms in Asians, three [23,24,25] in “Caucasians”, two [26,27] in mixed, and one [28] in African ethnicities. The source of controls was hospital-based in four [13,21,23,24] and population-based in four other articles [20,25,26,27]. All studies reported rs13041247 *MAFB* polymorphism involving 3082 NSCL/P patients and 4104 controls. Moreover, three studies reported *MAFB* rs11696257 polymorphism involving 845 NSCL/P patients and 927 controls. The *p*-value for HWE in controls of all studies was more than 0.05, and several genotyping methods were employed in the studies. In one of the studies [28], two cases had a family history.

### 3.3. Meta-Analysis Results

Figure 2 shows the pooled analysis for the risk of NSCL/P related to rs13041247 *MAFB* polymorphism based on five genetic models. The results identified that the pooled ORs of C vs. T, CC vs. TT, TC vs. TT, TC + CC vs. TT, and CC vs. TT + TC were 0.88 (95% CI: 0.75, 1.02; *p* = 0.09; I^2^ = 78% (P_h_ or P _heterogeneity_ < 0.00001)), 0.68 (95% CI: 0.48, 0.97; *p* = 0.03; I^2^ = 80% (P_h_ < 0.00001)), 0.81 (95% CI: 0.69, 0.95; *p* = 0.009; I^2^ = 51% (P_h_ = 0.02)), 0.79 (95% CI: 0.65, 0.95; *p* = 0.01; I^2^ = 70% (P_h_ = 0.0003)), and 0.82 (95% CI: 0.62, 1.07; *p* = 0.15; I^2^ = 74% (P_h_ < 0.0001)), respectively. Therefore, the risk of NSCL/P could be related to the CC and TC genotypes of rs13041247 *MAFB* polymorphism.

Figure 3 presents the pooled analysis of the risk of NSCL/P related to rs11696257 *MAFB* polymorphism based on five genetic models. The pooled ORs in the models of a vs. A, aa vs. AA, Aa vs. AA, Aa + aa vs. AA, and aa vs. AA + Aa were 0.90 (95% CI: 0.69, 1.19; *p* = 0.46; I^2^ = 72% (P_h_ = 0.03)), 0.70 (95% CI: 0.32, 1.54; *p* = 0.37; I^2^ = 80% (P_h_ = 0.007)), 0.95 (95% CI: 0.78, 1.17; *p* = 0.64; I^2^ = 0% (P_h_ = 0.44)), 0.93 (95% CI: 0.77, 1.13; *p* = 0.49; I^2^ = 39% (P_h_ = 0.20)), and 0.73 (95% CI: 0.37, 1.44; *p* = 0.37; I^2^ = 76% (P_h_ = 0.01)), respectively. Therefore, the risk of NSCL/P could not be related to rs11696257 *MAFB* polymorphism.

### 3.4. Subgroup Analysis

The results of subgroup analysis for the risk of NSCL/P related to rs13041247 *MAFB* polymorphism are shown in Table 2. Among the analyses, only one analysis (source of controls) showed a significant reduction in the risk of NSCL/P related to this polymorphism and other analyses did not identify any significant risk. It was found that the C allele and CC and TC genotypes in rs13041247 were associated with a decreased risk of NSCL/P compared with the T allele or TT genotype and also TC + CC could also decrease the risk of NSCL/P when compared to TT genotype. Therefore, C allele could potentially decrease the risk of NSCL/P in population-based studies.

### 3.5. Quality Assessment

The total score of quality of all articles included in this meta-analysis is shown in Table 3 that nine studies had score ≥7, high quality.

### 3.6. Sensitivity Analysis

Sensitivity analyses (one study removed and cumulative analyses) on previous analyses did not change the pooled results. Therefore, previous results had optimal stability. In addition, leave-out-one-study analysis on the quality of the studies deleting one study with low quality [22] showed that there was no considerable shift or alternation in heterogeneities and *p*-values of previous results.

### 3.7. Meta-Regression

Meta-regression analysis was performed for finding potential and possible sources of heterogeneity between rs13041247 polymorphism and susceptibility to NSCL/P (Table 4). The results showed that the year of publication, sample size, and *p*-value of the HWE were not the reasons for heterogeneity.

### 3.8. Publication Bias

Figure 4 shows the results of the funnel plot of the risk of NSCL/P related to rs13041247 and rs11696257 *MAFB* polymorphisms. The Begg’s and Egger’s tests did not reveal any publication bias across the studies in each analysis (*p* > 0.05).

## 4. Discussion

NSCL/P is a complex congenital anomaly to present both clinical and genetic heterogeneities [27]. This meta-analysis evaluated rs13041247 and rs11696257 polymorphisms of *MAFB* and the risk of NSCL/P. The rs11696257 polymorphism was not associated with the risk of NSCL/P, but the CC and TC genotypes rs13041247 polymorphism was associated with a significant decline in the risk of NSCL/P and had a protective role against NSCL/P with a heterogeneity of 80% for CC genotype and 51% for TC genotype. This significant decline in the risk of NSCL/P was observed in population-based studies with lower heterogeneities for the C allele and CC genotype (57% and 64%, respectively) or even lack of heterogeneity the TC genotype.

Out of 10 articles [13,20,21,22,23,24,25,26,27,28] including eleven studies reporting rs13041247 polymorphism in the present meta-analysis, three studies [13,20,22], four studies [13,20,22,24], and two studies [13,22] showed the C allele, the CC genotype, and the TC genotype to be associated with a significant decline in the risk of NSCL/P, respectively. Moreover, one study [21] showed the C allele and the CC genotype to be associated with a significant increase in the risk of NSCL/P. In addition, the TC + CC genotype was associated with a significant decline in the risk of NSCL/P in one study [1]. Out of three studies reporting rs11696257 polymorphism [21,24], one study [24] showed that the a allele and aa genotype were associated with a significant reduction in the risk of NSCL/P. One of the important reasons to show the difference in the results between these studies and our meta-analysis is that the present meta-analysis due to low studies reported had high heterogeneity. Another reason can be different genotyping methods in this meta-analysis. Therefore, further studies with more volume of participants are needed in the future. However, the association between the risk of NSCL/P and rs13041247 polymorphism in some studies and also this meta-analysis shows that the role of this polymorphism in NSCL/P patients cannot be ignored and well-designed studies in the future can be done to confirm the association with more power and accuracy. This association is robust across population-based studies in the present meta-analysis. In addition, the meta-regression likely lacked statistical power to detect effects of the three methodological variables (year of publication, sample size, and *p*-value of the HWE in controls), and also, there was no regular pattern when we examined the forest plots ordered by these variables. The expression of The *MAFB* mRNA and protein in the craniofacial neuroectoderm and the mesoderm are derived from the neural crest between embryonic days 13.5 and 14.5 [29]. This indicates that this gene may play a role in the development of NSCL/P [21]. *MAFB* has been shown as a candidate gene in NSCL/P pathogenesis according to genome-wide association study (GWAS) findings [30]. One study [31] on 298 case-parents trios reported that at rs13041247 polymorphism were associated with NSCL/P trios from the Western Han Chinese population and genotypic transmission-disequilibrium test analysis further confirmed this [31]. Another study on 1908 case-parent trios in a consortium drawing cases from several countries showed that polymorphisms of *MAFB* on 20q12 were not associated with NSCL/P achieved genome-wide significance [29]. In addition, the gene-gene interaction analysis showed interactions between *MAFB* and other genes in both European and Asian populations with NSCL/P [32].

The information regarding *MAFB* is scarce in the literature, especially in human models. Some studies [33,34,35] reported that the *MAFB* gene promotes the differentiation of some cells, such as phagocytic, *β*-islet, bone, and glomerular epithelial cells. In addition, this gene has been related to serum lipid concentrations, and also, the ischemic heart disease and stroke risk, which means that *MAFB* gene interacts with higher BMI, hypertension, and diabetes [36]. Therefore, genetic and environmental factors can interact with *MAFB* gene. Thus, the gene–gene and gene–environment interactions should be considered when assessing the association of polymorphisms of this gene with the risk of NSCL/P and other diseases.

This meta-analysis had several limitations such as (1) small number of studies available on this topic, (2) various genotyping methods, (3) high heterogeneity across the studies, (4) the Begg’s and Egger’s tests have low power, so it is difficult to exclude publication bias, and (5) the multiplicity of testing in the methods. However, the high quality of most studies and stability of the results were the strengths of this meta-analysis.

## 5. Conclusions

The results of the present meta-analysis show that the risk of NSCL/P is related to rs13041247 polymorphism, not rs11696257 polymorphism of *MAFB*. The CC and TC genotypes are significantly associated with the decreased risk of NSCL/P. In addition, a significant association was noted between rs13041247 polymorphism and a reduction in risk of NSCL/P in population-based studies. Well-designed studies are required to assess the interaction of *MAFB* gene and other genes with environmental factors in different ethnic groups.

## Figures and Tables

**Figure 1 ijerph-16-02792-f001:**
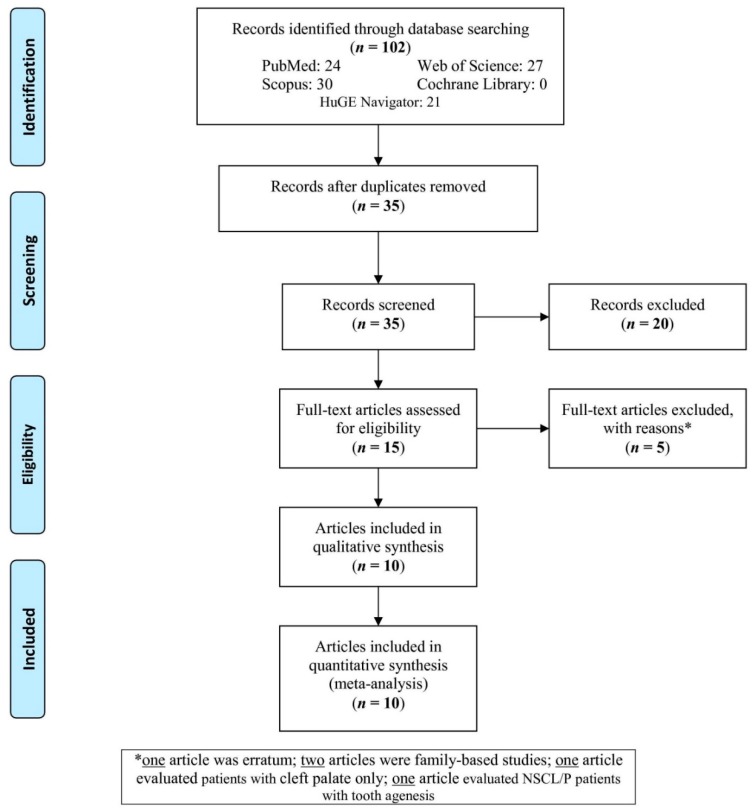
Flow-chart of the article selection.

**Figure 2 ijerph-16-02792-f002:**
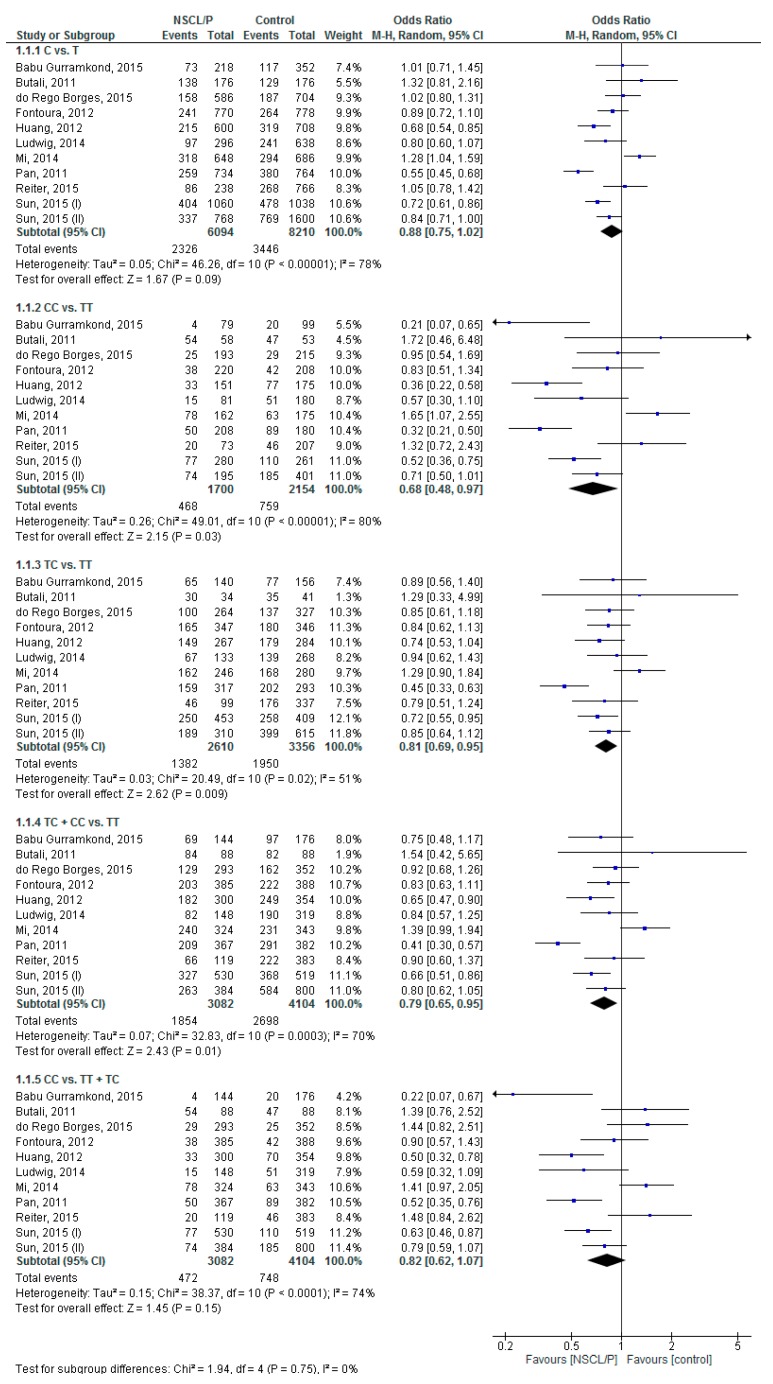
Forest plot of the odds ratio of the risk of non-syndromic cleft lip/palate related to rs13041247 *MAFB* polymorphism.

**Figure 3 ijerph-16-02792-f003:**
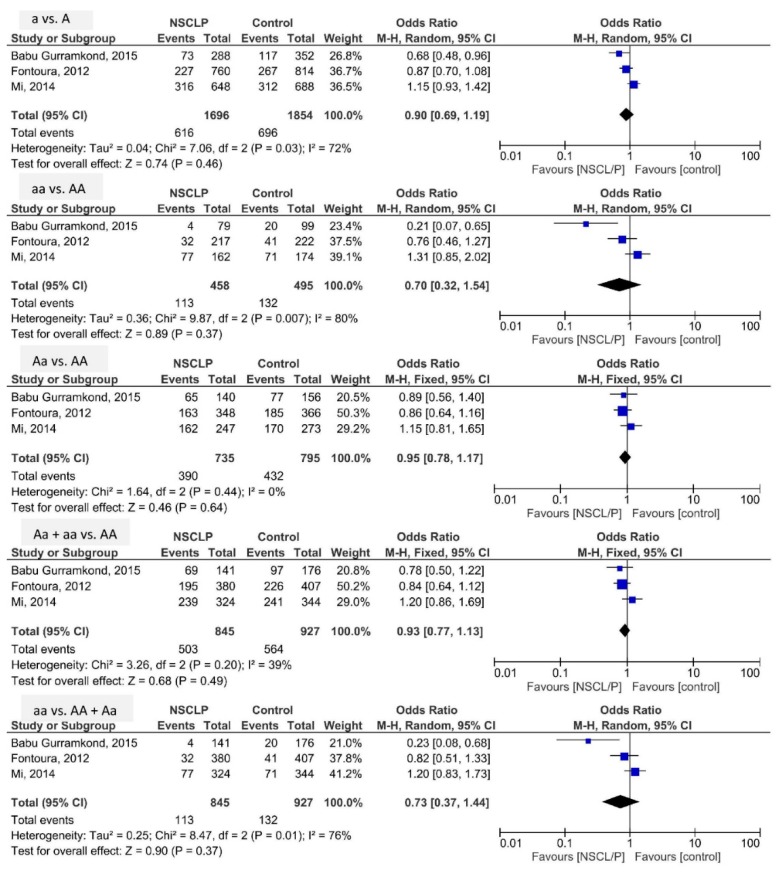
Forest plot of the odds ratio of the risk of non-syndromic cleft lip/palate related to rs11696257 *MAFB* polymorphism.

**Figure 4 ijerph-16-02792-f004:**
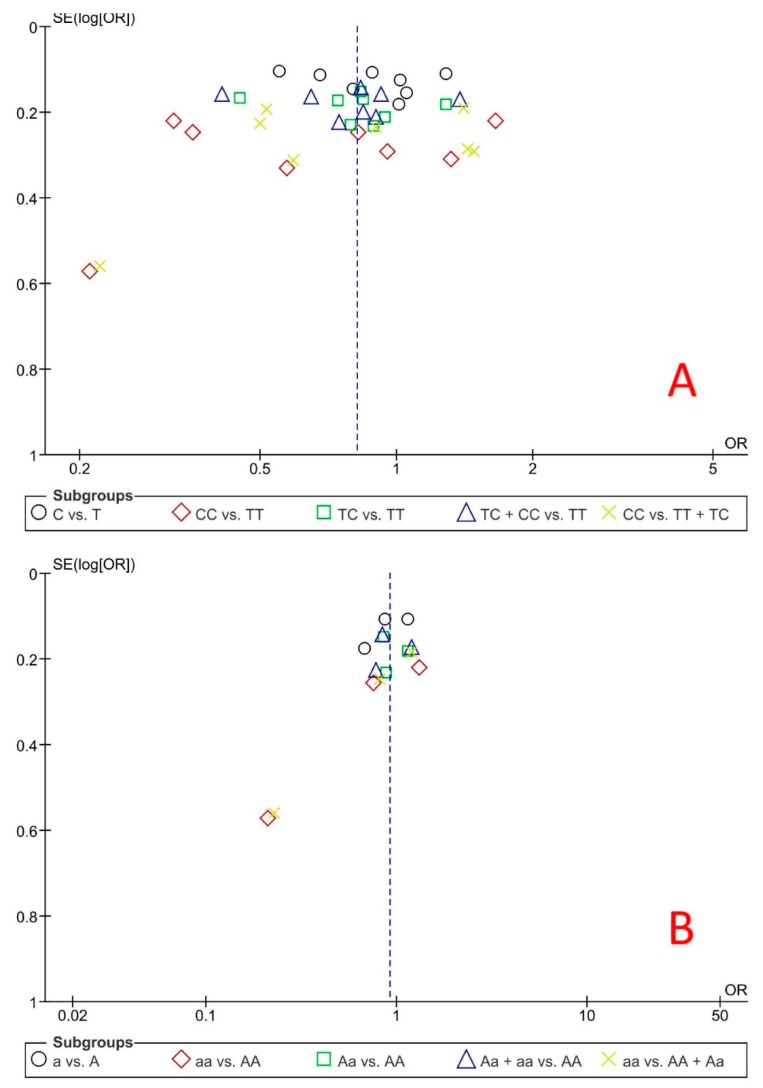
Funnel plot of the risk of non-syndromic cleft lip/palate related to (**A**) rs13041247 and (**B**) rs11696257 *MAFB* polymorphisms based on five genetic models (SE: standard errors vs. OR: Odds ratio).

**Table 1 ijerph-16-02792-t001:** Basic characteristics of eleven studies included in this meta-analysis.

First Author, Publication Year	Ethnicity	Source of Controls	NSCL/P		Control		Genotyping Method	*p*-Value for HWE in Controls
			TT/TC/CC	AA/Aa/aa	TT/TC/CC	AA/Aa/aa		
Butali, 2011 [28]	African	PB	4/30/54	-	6/35/47	-	PCR	0.880
Pan, 2011 [13]	Asian	HB	158/159/50	-	91/202/89	-	PCR	0.260
Fontoura, 2012 [23]	“Caucasian”	HB	182/165/38	185/163/32	166/180/42	181/185/41	TaqMan	0.511/0.530
Huang, 2012 [20]	Asian	PB	118/149/33	-	105/19/70	-	Mass spectrometry	0.689
Ludwig, 2014 [26]	Mixed	PB	66/67/15	-	129/139/51	-	PCR-RFLP	0.192
Mi, 2014 [21]	Asian	HB	84/162/78	85/162/77	112/168/63	103/170/71	Mini-sequencing	1.000/0.956
Babu Gurramkond, 2015 [24]	“Caucasian”	HB	75/65/4	75/65/4	79/77/20	79/77/20	KASPar	0.850/0.850
do Rego Borges, 2015 [27]	Mixed	PB	164/100/29	-	190/137/25	-	Real-Time PCR	0.964
Reiter, 2015 [25]	“Caucasian”	PB	53/46/20	-	161/176/46	-	PCR	0.843
Sun, 2015 (I) * [22]	Asian	PB	203/250/77	-	151/258/110	-	Affymetrix Genome-Wide	0.991
Sun, 2015 (II) ** [22]	Asian	PB	121/189/74	-	216/399/185	-	Affymetrix Genome-Wide	0.977

Abbreviations: HB, hospital-based; PB, population-based; HWE, Hardy–Weinberg equilibrium; PCR: polymerase chain reaction; RFLP: restriction fragment length polymorphism; KASPar, allele-specific amplification followed by fluorescence detection; NSCL/P, non-syndromic cleft/palate. * Huaxi Cohort. ** Nanjing Cohort.

**Table 2 ijerph-16-02792-t002:** Subgroup analysis of the risk of non-syndromic cleft lip/palate related to rs13041247 *MAFB* polymorphism.

Subgroup Analysis (*n*)	C vs. T	CC vs. TT	TC vs. TT	TC + CC vs. TT	CC vs. TT + TC
	OR (95% CI), I^2^ (%), P_h_	OR (95% CI), I^2^ (%), P_h_	OR (95% CI), I^2^ (%), P_h_	OR (95% CI), I^2^ (%), P_h_	OR (95% CI), I^2^ (%), P_h_
Overall (11)	0.88 (0.75, 1.02), 78, <0.00001	**0.68 (0.48, 0.97), 80, <0.00001**	**0.81 (0.69, 0.95), 51, 0.02**	**0.79 (0.65, 0.95), 70, 0.0003**	0.82 (0.62, 1.07), 74, <0.0001
Ethnicity					
Asian (5)	0.78 (0.60, 1.01), 88, <0.00001	0.59 (0.35, 1.01), 88, <0.00001	0.76 (0.56, 1.03), 78, 0.001	0.72 (0.51, 1.03), 86, <0.0001	0.72 (0.51, 1.02), 79, 0.0009
“Caucasian” (3)	0.95 (0.81, 1.11), 0, 0.62	0.70 (0.32, 1.56), 75, 0.02	0.84 (0.67, 1.04), 0, 0.94	0.83 (0.68, 1.02), 0, 0.83	0.77 (0.34, 1.73), 78, 0.01
Mixed (2)	0.92 (0.76, 1.11), 34, 0.22	0.76 (0.49, 1.16), 24, 0.25	0.88 (0.68, 1.14), 0, 0.69	0.86 (0.69, 1.06), 0, 0.75	0.93 (0.39, 2.22), 77, 0.04
Source of controls					
Hospital-based (4)	0.89 (0.60, 1.31), 91, <0.00001	0.59 (0.25, 1.42), 91, <0.00001	0.81 (0.52, 1.25), 84, 0.0004	0.77 (0.47, 1.28), 89, <0.00001	0.70 (0.38, 1.30), 84, 0.0002
Population-based (7)	**0.85 (0.74, 0.99), 57, 0.03**	**0.69 (0.49, 0.97), 64, 0.01**	**0.80 (0.70, 0.92), 0, 0.91**	**0.78 (0.68, 0.88), 0, 0.47**	0.86 (0.63, 1.17), 69, 0.004
Genotyping method					
PCR-based (5)	0.89 (0.65, 1.21), 83, 0.0001	0.76 (0.41, 1.43), 79, 0.0007	0.74 (0.53, 1.03), 63, 0.03	0.77 (0.52, 1.14), 77, 0.002	0.96 (0.59, 1.58), 77, 0.002
Others (6)	0.88 (0.73, 1.05), 78, 0.0005	0.63 (0.40, 1.01) 83, <0.0001	0.85 (0.74, 0.97), 31, 0.20	0.81 (0.66, 1.01), 65, 0.01	0.73 (0.51, 1.03), 76, 0.001

Abbreviations: OR, odds ratio; CI, confidence interval, P_h_, P_heterogeneity_. Bold numbers indicate significant differences (*p* < 0.05).

**Table 3 ijerph-16-02792-t003:** Quality assessment scores for the studies included in this meta-analysis.

First Author (year)	Selection	Comparability	Exposure	Total Points
**Butali, 2011 [28]**	***	*	***	7
**Pan, 2011 [13]**	***	**	***	8
**Fontoura, 2012 [23]**	**	*	**	5
**Huang, 2012 [20]**	****	*	***	8
**Ludwig, 2014 [26]**	****	-	***	7
**Mi, 2014 [21]**	***	**	***	8
**Babu Gurramkond, 2015 [24]**	***	**	***	8
**do Rego Borges, 2015 [27]**	****	-	***	7
**Reiter, 2015 [25]**	***	*	***	7
**Sun, 2015 [22]**	****	*	***	8

Each asterisk means one point.

**Table 4 ijerph-16-02792-t004:** Meta-regression analyses on year, sample size, and *p*-value of the HWE in controls for finding a potential source of heterogeneity between rs13041247 polymorphism and susceptibility to non-syndromic cleft lip/palate.

Variable	C vs. T	CC vs. TT	TC vs. TT	TC + CC vs. TT	CC vs. TT + TC
Year of publication	*r* = 0.153 (*p* = 0.652)	*r* = 0.070 (*p* = 0.837)	*r* = 0.026 (*p* = 0.939)	*r* = 0.068 (*p* = 0.842)	*r* = 0.067 (*p* = 0.844)
sample size	*r* = −0.531 (*p* = 0.093)	*r* = −0.305 (*p* = 0.362)	*r* = −0.447 (*p* = 0.168)	*r* = −0.465 (*p* = 0.149)	*r* = −0.179 (*p* = 0.599)
*p*-value of the HWE in controls	*r* = 0.548 (*p* = 0.081)	*r* = 0.408 (*p* = 0.212)	*r* = 0.411 (*p* = 0.209)	*r* = 0.439 (*p* = 0.176)	*r* = 0.434 (*p* = 0.182)

*r* means correlation coefficient.

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
