# Peer review of "Polymorphic Variants of V-Maf Musculoaponeurotic Fibrosarcoma Oncogene Homolog B (rs13041247 and rs11696257) and Risk of Non-Syndromic Cleft Lip/Palate: Systematic Review and Meta-Analysis"

_ijerph, 2019, doi:10.3390/ijerph16152792_

Round 1

Reviewer 1 Report

The authors report a systematic review and meta-analysis of the association between non-syndromic cleft lip/palate and two polymorphic variants of the MAFB gene, concluding that there is no overall association.

Comments

I think I understand from the paper that only studies that included both non-syndromic cleft lip (with or without associated cleft palate) were considered, but not non-syndromic cleft palate. I note that one study was excluded because it only related to cleft palate. If this is correct, I am surprised at this decision, as it would add to the paper to undertake systematic review/meta-analysis by type of cleft. Also, the volume of evidence on cleft palate tends to be less than that for cleft lip, so meta-analysis is particularly valuable for increasing statistical power (if the studies are not too different in design).

As it stands, the rationale for having undertaken this systematic review and meta-analysis is not strong. It is open to the criticism made of meta-analyses of candidate genetic association studies made by Ioannidis (Milbank Q. 2016 Sep;94(3):485-514. doi: 10.1111/1468-0009.12210.). The biologic rationale provided in the introduction is not clearly relevant to orofacial development. I had to go back to some original papers to understand that SNPs tagging MAFB had been identified and replicated in a genome wide association study (Beaty et al., ref 28), in which studies of gene expression of Mafb in a mouse model had detected increased expression in tissues relevant to oral cleft development. I would suggest that this is clearly explained in the introduction to the paper.

Materials and methods - the PRISMA protocol is a guideline for reporting systematic reviews and meta-analyses, NOT for the conduct of such studies.

Search strategy - For this topic, I was surprised that the Cochrane Library was considered, but not EMBASE or the HuGE Navigator (https://phgkb.cdc.gov/PHGKB/hNHome.action).

Eligibility criteria - The authors state that family-based studies were excluded. Did this mean that case-parent trio studies were excluded. Many genetic association studies of congenital anomalies, including orofacial clefts, have been based on case-parent trio studies (and sometimes there is a hybrid case-control case-parent trio design). Case-parent trio studies can be analysed to yield relative risks (very similar magnitudes of effect to odds ratios provided the condition is uncommon, like non-syndromic orofacial clefts - see Ahsan et al. Int J Epidemiol. 2002 Jun;31(3):669-78.) I did not understand why your search did not pick up the study of  Butali et al Cleft Palate Craniofac J2011 Nov;48(6):646-53. doi: 10.1597/10-133. (2 of cases had family history; FBAT analysis was used, but there are raw data on cases and controls) or that of Sun et al Nat Commun. 2015 Mar 16;6:6414. doi: 10.1038/ncomms7414.

Quality assessment - why no verification of quality evaluation?

Qualitative synthesis - what is rationale for 5 genetic models? Usual to consider additive, dominant and recessive. Is this multiplicity of models making your study vulnerable to multiple testing? Also see Minelli C et al. Int J Epidemiol. 2005 Dec;34(6):1319-28.

Statistical analysis - There is a great deal of literature suggesting that it is inappropriate to make a decision about random vs fixed effects meta-analysis based on results of a so-called heterogeneity test (these tests are not statistically powerful). See for example https://handbook-5-1.cochrane.org/chapter_9/9_5_4_incorporating_heterogeneity_into_random_effects_models.htm

Study characteristics - Use of the term "Caucasian". I know that it is widely used, and is in the upper-reviewed publications included in the thesis. However, many are critical of this term - see http://www.nytimes.com/2013/07/07/sunday-review/has-caucasian-lost…ing.html?nl=todaysheadlines&emc=edit_th_20130707&pagewanted=print  https://ajph.aphapublications.org/doi/pdf/10.2105/AJPH.88.9.1303  https://www.ncbi.nlm.nih.gov/pubmed/29396816

Publication bias - Note that Begg test has low statistical power. I'm not clear that it was appropriate to use the Egger test. See https://wiki.joannabriggs.org/display/MANUAL/3.3.11+Publication+bias

Discussion - have gone to considerable care to undertake meta-analysis, why resort to vote-counting in discussion?

Author Response

Q1. the PRISMA protocol is a guideline for reporting systematic reviews and meta-analyses, NOT for the conduct of such studies.

Answer: Please see below articles and several other studies about this type of studies and PRISMA. “A Systematic Review of Single Nucleotide Polymorphisms Associated With Metabolic Syndrome in Children and Adolescents”, “Associations of tumor necrosis factor-α polymorphisms with the risk of colorectal cancer: a meta-analysis”, “A meta-analysis: Is there any association between MiR-608 rs4919510 polymorphism and breast cancer risks?”, “Association between Genetic Polymorphisms in Interleukin Genes and Recurrent Pregnancy Loss – A Systematic Review and Meta-Analysis”, “No Correlation between TIMP2 -418 G>C Polymorphism and Increased Risk of Cancer: Evidence from a Meta-Analysis”.

Q2: For this topic, I was surprised that the Cochrane Library was considered, but not EMBASE or the HuGE Navigator.

Answer: Thanks for your valuable comment. I searched the studies with this topic. I found different databases for searching.  These databases are selected based on the policy of our university for meta-analyses and with the advice of the statisticians and we do not have access to Embase (account of Embase by our university has not purchased).

Q3: The authors state that family-based studies were excluded. Did this mean that case-parent trio studies were excluded. Many genetic association studies of congenital anomalies, including orofacial clefts, have been based on case-parent trio studies (and sometimes there is a hybrid case-control case-parent trio design). Case-parent trio studies can be analysed to yield relative risks (very similar magnitudes of effect to odds ratios provided the condition is uncommon, like non-syndromic orofacial clefts - see Ahsan et al. Int J Epidemiol. 2002 Jun;31(3):669-78.)

Answer: We changed criteria.

Q4: I did not understand why your search did not pick up the study of Butali et al Cleft Palate Craniofac J. 2011 Nov;48(6):646-53. doi: 10.1597/10-133. (2 of cases had family history; FBAT analysis was used, but there are raw data on cases and controls) or that of Sun et al Nat Commun. 2015 Mar 16;6:6414. doi: 10.1038/ncomms7414.

Answer: First of all, one of problems of the study of Butali et al. that was lack of prevalence of genotypes and HWE for controls was 0.105. Because HWE is faraway 1.00, calculating genotype prevalence of controls from alleles, there was a pretty much error. However, this study is a Family-Based CaseControl study and this type of studies is different with unrelated case-control studies. Please see “Evangelos Evangelou, Thomas A Trikalinos, Georgia Salanti, John P. A Ioannidis.  PLoS Genet. 2006 Aug; 2(8): e123. doi: 10.1371/journal.pgen.0020123”. Also, 2 of cases had family history and FBAT analysis was used.  The study of Sun et al. was a Genome-wide association study that the authors didn’t report frequency of alleles and genotypes’ prevalence (not sufficient data). This study was deleted with reason given in below of Figure 1.  

Q5: why no verification of quality evaluation.

Answer: The Newcastle-Ottawa scale (NOS) was applied for the quality assessment of articles.

Q6: What is rationale for 5 genetic models? Usual to consider additive, dominant and recessive. Is this multiplicity of models making your study vulnerable to multiple testing? Also see Minelli C et al. Int J Epidemiol. 2005 Dec;34(6):1319-28.

Answer: Reporting a study based on 5 genetic models has been shown in a lot of articles. We searched that a study with 5 genetic models has more complete analysis to evaluate susceptibility of polymorphisms.      

Q6: There is a great deal of literature suggesting that it is inappropriate to make a decision about random vs fixed effects meta-analysis based on results of a so-called heterogeneity test (these tests are not statistically powerful)

Answer: We checked heterogeneity in previous meta-analyses. We figure out a result about it that it has been written in our meta-analysis.  

Q7: Use of the term "Caucasian". I know that it is widely used, and is in the upper-reviewed publications included in the thesis.

Answer: Thanks for your valuable comment. As in below article: “BMP4 rs17563 polymorphism and nonsyndromic cleft lip with or without cleft palate A meta-analysis” and also “https://en.wikipedia.org/wiki/Caucasian_race”.

Q8: Note that Begg test has low statistical power. I'm not clear that it was appropriate to use the Egger test.

Answer: In CMA software for each analysis with almost 3 articles, the results of Begg’s and Egger’s test obtain. In most articles, I found the results of both tests have been written together.  

Q9: have gone to considerable care to undertake meta-analysis, why resort to vote-counting in discussion?

Answer: We did not understand your comment very accurately. Nevertheless, we described the results of each study included in the meta-analysis and then described our results of meta-analysis to show effect of the results of each individual study compared to a pooled analysis. If it is necessary that we have a more description, we can do.

Reviewer 2 Report

  There is a well prepared meta-analysis based on eight studies.

  However, there is hard to accept that “lack of publication bias” was recognized. IN general, there is always an publication bias, it can be low and bad no statistically significant effect.

  It was recognized that “ some recent meta-analyses showed the association of genetics with the development of NSCL/P”. The present meta-analysis found no association. Therefore, under Discussion, some explanation why previously an association was reported, is needed.

  The quality of Fig.4 should be improved (the color of graphical characters should be more intensive because some characters are not well recognizable)

Author Response

Q1: Therefore, under Discussion, some explanation why previously an association was reported, is needed.

Answer: One of important reasons to show difference of the results between these studies and our meta-analysis is that the present meta-analysis due to low studies reported had a high heterogeneity. Another reason can be different genotyping methods in this meta-analysis. Therefore, it is needed further studies with more volume of participates in future.   

Q2: The quality of Fig.4 should be improved (the color of graphical characters should be more intensive because some characters are not well recognizable)

Answer: We changed figure with high quality.

Round 2

Reviewer 1 Report

Thank you for having considered the comments on the previous submission. However, I do not think that the responses are satisfactory.

It may be the case that other authors have asserted that they have used PRISMA as guideline for conduct of systematic reviews and meta-analyses, but in fact PRISMA is a reporting guideline - just because others say that they have done this doesn't mean that they are correct! It would OK for you state that your reporting of your study followed PRISMA guidance.

Fair enough point about not using EMBASE, but why not HuGE Navigator https://phgkb.cdc.gov/PHGKB/hNHome.action  ?

In response to my earlier point about case-parent studies, and explanation that risk estimates from them could be included along with magnitudes of association derived from case-control studies, you decided to exclude them. This seems a very peculiar decision. There is a lack of evidence, and I think it would be valuable to check if there are such studies, and investigate effect on overall estimate of association through sensitivity analyses.

The Butali study includes cases and controls, and numbers are presented of matched case and control probands with complete genotype and maternal environment factor data (Table 4). The frequencies related to proportions with one and two alleles, so it should be possible to calculate Ns for genotypes. Mothers of cases and controls are also included, and a very small number of trios. The p value for HWE in controls is 0.105, which does not indicate significant departure from equilibrium, and you say in your paper that you had 0.05 threshold. It is correct that two cases had family history, so that might inflate association, but you could address that through sensitivity analysis. Its not really a family based study in the sense of Evangelou et al. 

Regarding the Sun et al study, allele and genotype frequency are presented in Supplementary Table 5 - for GWASI, GWAS II, Validation Ia, Ib and Ic, and Validation II, so I do not agree with your rationale for exclusion.

Thank you for adding assessment of studies using NOS. However, you have not assessed the implications of this for the results of the meta-analyses. For example, how does it fit with your leave-out-one-study analysis, and did you consider in your meta-regression?

OK, so you are using five models to have a "more complete analysis", but this does raise an issue of multiplicity of testing. You should comment on this.

I did not find your answer to Q6 convincing. You are including observational studies, so a priori a random effects analysis likely to be more appropriate than fixed effects. See https://bmcmedresmethodol.biomedcentral.com/articles/10.1186/s12874-018-0495-9

I would suggest that you at least put "Caucasian" in quotation marks. See  http://www.nytimes.com/2013/07/07/sunday-review/has-caucasian-lost…ing.html?nl=todaysheadlines&emc=edit_th_20130707&pagewanted=print  https://ajph.aphapublications.org/doi/pdf/10.2105/AJPH.88.9.1303  https://www.ncbi.nlm.nih.gov/pubmed/29396816

I accept that CMA gives you the results of applying the Begg and Egger tests, but in the discussion of publication bias, important to note that these tests ave low power, so it is difficult to exclude publication bias

With regard to Q9, I was surprised that you did not summarize your study results in terms of the magnitude of association and associated estimate of heterogeneity. Instead you present, x studies say this, y studies say that, which does not take account of study characteristics (number of participants, evidence of bias etc). Please reconsider

Author Response

Dear Reviewer

Q1: It may be the case that other authors have asserted that they have used PRISMA as guideline for conduct of systematic reviews and meta-analyses, but in fact PRISMA is a reporting guideline - just because others say that they have done this doesn't mean that they are correct! It would OK for you state that your reporting of your study followed PRISMA guidance.

Answer: We changed it based on your comment.

Q2: Fair enough point about not using EMBASE, but why not HuGE Navigator https://phgkb.cdc.gov/PHGKB/hNHome.action?

 Answer: We added HuGE Navigator.

Q3: In response to my earlier point about case-parent studies, and explanation that risk estimates from them could be included along with magnitudes of association derived from case-control studies, you decided to exclude them. This seems a very peculiar decision. There is a lack of evidence, and I think it would be valuable to check if there are such studies, and investigate effect on overall estimate of association through sensitivity analyses.

Answer: We deleted case-parent studies and added “the studies reporting parents who have children with NSCL/P as cases and/or parents without NSCL/P children as controls”

Q4: The Butali study includes cases and controls, and numbers are presented of matched case and control probands with complete genotype and maternal environment factor data (Table 4). The frequencies related to proportions with one and two alleles, so it should be possible to calculate Ns for genotypes. Mothers of cases and controls are also included, and a very small number of trios. The p value for HWE in controls is 0.105, which does not indicate significant departure from equilibrium, and you say in your paper that you had 0.05 threshold. It is correct that two cases had family history, so that might inflate association, but you could address that through sensitivity analysis. Its not really a family based study in the sense of Evangelou et al.

Answer: We added Butali’s study.

Q5: Regarding the Sun et al study, allele and genotype frequency are presented in Supplementary Table 5 - for GWASI, GWAS II, Validation Ia, Ib and Ic, and Validation II, so I do not agree with your rationale for exclusion.

Answer: We added Sun et al. study.

Q6: Thank you for adding assessment of studies using NOS. However, you have not assessed the implications of this for the results of the meta-analyses. For example, how does it fit with your leave-out-one-study analysis, and did you consider in your meta-regression?

Answer: We described the result of leave-out-one-study analysis.

Q7: OK, so you are using five models to have a "more complete analysis", but this does raise an issue of multiplicity of testing. You should comment on this.

Answer: We commented in text.

Q8: did not find your answer to Q6 convincing. You are including observational studies, so a priori a random effects analysis likely to be more appropriate than fixed effects. See https://bmcmedresmethodol.biomedcentral.com/articles/10.1186/s12874-018-0495-9

Answer: A lot of studies reporting observational studies in meta-analysis used fixed effects analysis, if P-value >0.1 (I2<50%) that in this journal “Int. J. Environ. Res. Public Health”, there were some articles about this issue. We used them for description. However, your valuable comment has been used in some studies. If it is necessary, we can change it based your comment.

Q9: I would suggest that you at least put "Caucasian" in quotation marks. See  http://www.nytimes.com/2013/07/07/sunday-review/has-caucasian-lost…ing.html?nl=todaysheadlines&emc=edit_th_20130707&pagewanted=print  https://ajph.aphapublications.org/doi/pdf/10.2105/AJPH.88.9.1303  https://www.ncbi.nlm.nih.gov/pubmed/29396816

Answer: we added quotation marks.

Q10: I accept that CMA gives you the results of applying the Begg and Egger tests, but in the discussion of publication bias, important to note that these tests ave low power, so it is difficult to exclude publication bias.

Answer: We added to limitations.

Q11: With regard to Q9, I was surprised that you did not summarize your study results in terms of the magnitude of association and associated estimate of heterogeneity. Instead you present, x studies say this, y studies say that, which does not take account of study characteristics (number of participants, evidence of bias etc).

Answer: We added descriptions in discussion. Our aim of adding ”x studies say this, y studies say” was that show the results of studies included in the meta-analysis was different and we can’t reject the association between the polymorphism simply and it needs  future studies for more precise explanation.

Round 3

Reviewer 1 Report

Appreciate more focused responses to the points raised on the last version

Some suggestions:

Change "This meta-analysis followed the PRISMA guidance" to "This meta-analysis is reported according to the PRISMA guidance"

I accept your choice as investigators to exclude case-parent studies. All you have done is to change the wording about this exclusion. Nevertheless, you should at least put a sentence or two in your Discussion documenting the decision - you could note where the studies of this type were done, their size (i.e.number of complete and partial trios included in analysis) and whether the association was in the same direction (and magnitude if reported) as you found in the meta-analysis

I note that you have mentioned multiplicity of testing in the methods, but  you should also consider this in the discussion of limitations. 

The text in lines 229-232 is vague - You state "However, the association between the risk of NSCL/P and rs13041247 polymorphism in some studies and also this meta-analysis shows that the role of this polymorphism in future studies can be noticed. The association was stronger about TC genotype, especially in population-based studies that had lack of heterogeneity." It looks to me as if the association is robust across ethnicities, but is more marked in hospital based studies and in studies that did not use PCR methods for genotyping.

The meta-regression likely lacked statistical power to detect effects of the three methodological variables considered, but was there any pattern apparent when you examined the forest plots ordered by these variables? You could add consideration of that in to point 4

Author Response

Dear

Q1: Change "This meta-analysis followed the PRISMA guidance" to "This meta-analysis is reported according to the PRISMA guidance".

Answer: We changed it.

Q2: I accept your choice as investigators to exclude case-parent studies. All you have done is to change the wording about this exclusion. Nevertheless, you should at least put a sentence or two in your Discussion documenting the decision - you could note where the studies of this type were done, their size (i.e.number of complete and partial trios included in analysis) and whether the association was in the same direction (and magnitude if reported) as you found in the meta-analysis.

Answer: We added your valuable comment to discussion and results.

Q3: I note that you have mentioned multiplicity of testing in the methods, but  you should also consider this in the discussion of limitations. 

Answer: We added it to limitations.

Q4: The text in lines 229-232 is vague - You state "However, the association between the risk of NSCL/P and rs13041247 polymorphism in some studies and also this meta-analysis shows that the role of this polymorphism in future studies can be noticed. The association was stronger about TC genotype, especially in population-based studies that had lack of heterogeneity." It looks to me as if the association is robust across ethnicities, but is more marked in hospital based studies and in studies that did not use PCR methods for genotyping.

Answer: We changed lines.

Q5: The meta-regression likely lacked statistical power to detect effects of the three methodological variables considered, but was there any pattern apparent when you examined the forest plots ordered by these variables? You could add consideration of that in to point 4.

Answer: We added it to point 4, whatever; this comment was almost unclear for us. In addition, there was no regular pattern when we examined the forest plots ordered by these variables. If it needs more descriptions, please guide us more.